# Efficacy and Safety of Nitrous Oxide (N_2_O) Inhalation Sedation Compared to Other Sedative Agents in Dental Procedures: A Systematic Review with Meta-Analysis

**DOI:** 10.3390/medicina61050929

**Published:** 2025-05-20

**Authors:** Francesca Piccialli, Marco Fiore, Roberto Giurazza, Fabrizio Falso, Vittorio Simeon, Paolo Chiodini, Diana Russo, Luigi Laino

**Affiliations:** 1Department of Women, Child and General and Specialized Surgery, University of Campania “Luigi Vanvitelli”, Via Luigi De Crecchio, 2, 80138 Naples, Italy; francesca.piccialli@studenti.unicampania.it (F.P.); roberto.giurazza@unicampania.it (R.G.); 2Department of Critical Care, AORN Ospedali dei Colli, Via Leonardo Bianchi, 80131 Naples, Italy; fabriziofalso@gmail.com; 3Medical Statistics Unit, Department of Public, Clinical and Preventive Medicine, University of Campania “Luigi Vanvitelli”, 80138 Naples, Italy; 4Multidisciplinary Department of Medical-Surgical and Odontostomatological Specialties, University of Campania “Luigi Vanvitelli”, 80138 Naples, Italy; diana.russo@unicampania.it (D.R.); luigi.laino@unicampania.it (L.L.)

**Keywords:** nitrous oxide, sedatives, dental procedures, patient safety, systematic review, meta-analysis

## Abstract

*Background and Objectives*: Dental procedures can be distressing, particularly for patients who are anxious or uncooperative. In such cases, effective sedation not only facilitates the clinician’s work but also enhances patient comfort and acceptance of dental care. Nitrous oxide (N_2_O) has been widely employed as a sedative agent in dental practice. This systematic review aimed to assess the efficacy and safety of N_2_O compared to alternative sedative agents and techniques in dental surgical procedures. *Materials and Methods:* This review protocol was prospectively registered on PROSPERO (CRD42020213429) and conducted in accordance with PRISMA guidelines. *Results:* A total of 1809 records were screened (1134 from Embase, 638 from PubMed, and 37 from CENTRAL). The meta-analysis focused on the following three primary outcomes: 1. Patient satisfaction: Eight studies comprising 422 participants (165 treated with N_2_O and 176 with other sedatives) were included. 2. Recall of the procedure: Five studies with a total of 288 patients (68 receiving N_2_O and 64 other agents) assessed patients’ ability to recall tooth extraction. 3. Successful completion of the procedure: Three studies involving 1578 patients (434 treated with N_2_O and 989 with alternative sedation methods) were analyzed. *Conclusions:* No statistically significant differences were observed between N_2_O and other sedative techniques across all outcomes evaluated. Safety could not be adequately assessed as none of the included studies systematically investigated this outcome. Further high-quality research is warranted to rigorously evaluate the safety profiles of various sedation strategies in dental surgery.

## 1. Introduction

For some individuals, both children and adults, dental treatments can be a frightening experience [1]. Therefore, managing anxious and uncooperative patients can be challenging during awake procedures, such as in dental practice [2]. Children’s anxiety towards dental procedures is a universal issue, but it varies worldwide in different cultures and countries, ranging from 3% to 43% among the different studies [3].

Providing anxiolysis through psychological techniques may not be constantly effective for all patients, sometimes requiring the use of pharmacological means of sedation. In some individuals, especially in uncooperative children, adequate sedation simplifies the dentist’s work, makes the dental procedure faster, and positively affects the patient’s comfort and attitude towards dentistry [4].

It is estimated that up to 20% of the population experiences different degrees of dental anxiety, which not only has an impact on the psyche of the individual but also significantly affects dental attendance, oral care, and general overall oral health, sometimes leading to carelessness and decreased quality of life [5]. The existence of a vicious cycle of dental fear, fewer dentist visits, delay of dental treatments, bigger oral health problems, need for more invasive procedures, and higher dental fear has been hypothesized [6]. Consequently, patients with higher degrees of dental anxiety more often experience oral health problems and exhibit a higher number of missing teeth [7,8]. Therefore, providing appropriate anxiolysis and adequate sedation to odontophobic patients is something necessary to allow their dental treatments and improve their oral health [5,9].

Dental procedures are generally performed in an outpatient setting, hence the ideal sedative agent for these procedures should be safe, providing adequate and efficient analgesia and sedation, while at the same time keeping the patient cooperative. It should also have a fast onset and a fast offset with no residual sedation, should cause no or minimal cardiorespiratory depression, and should have negligible and bearable side effects [4,10,11].

Several sedative agents and their combinations have been reported in the literature for dental procedures, such as inhalational agents, short-acting benzodiazepines, barbiturates, chloral hydrate, antihistamines, and opioids [12,13]. Different routes of administration are available for these drugs, such as oral, sublingual, rectal, intramuscular, inhalational, and intravenous, depending on the patient and the specific drug the anesthesiologist has chosen.

Anyways, in some very limited cases, general anesthesia is still necessary according to the surgeon’s preference, in cases of failure of other sedation techniques, with patients with severe disabilities, or in complex and long dental procedures that would not be suitable for sedation [14,15]. Anyways, compared to sedation, general anesthesia requires an inpatient facility and has higher costs and a higher rate of side effects and mortality [2,16,17].

One of the most common and oldest ways to achieve dental sedation is nitrous oxide/oxygen inhalation, which has been accounted by the American Academy of Pediatric Dentistry (AAPD) as a safe and efficient technique to provide analgesia, anxiolysis, and conscious sedation, and enhance cooperation between patient and dentist during dental interventions [18].

Nitrous oxide is a gaseous inhalational anesthetic, with a light sweet smell, that can be easily available in an outpatient setting, such as the dentist’s office, where it can be effectively administered via a nasal hood in a mixture with oxygen [19,20]. It has a very low blood–gas and fat–blood partition coefficient, reflecting a low degree of solubility in blood and adipose tissue, respectively. These physicochemical properties reflect the unique and favourable pharmacokinetic profile of this gas: during induction it is rapidly absorbed in the blood and the CNS and likewise it is rapidly and completely cleared off after its administration is interrupted [21]. Nitrous oxide has an extremely low potency as a monoanesthetic (MAC = 104), but subanesthetic concentrations can safely provide analgesia and relief of anxiety, causing euphoria and CNS depression, while at the same time causing minimal cardiorespiratory depression and not abolishing airway reflexes [22].

The precise mechanism of action of nitrous oxide is not completely known. The analgesic effect seems to be mediated by the release of endogenous opioid peptides (encephalins), which activate descending noradrenergic and serotoninergic analgesic pathways in the CNS [23]. The anxiolytic effect is mediated by the activation of the GABA_A_ receptor, probably through the benzodiazepine binding site [24,25].

The purpose of this systematic review is to compare the efficacy and safety of nitrous oxide (N_2_O) to other sedative agents and techniques used in dental surgical procedures.

## 2. Materials and Methods

After excluding the existence of systematic reviews on the same topic by searching the primary electronic registries (CENTRAL, PROSPERO, and the JBI Database of Systematic Reviews and Implementation Report), we registered the systematic review protocol on 7 December 2020 (PROSPERO: CRD42020213429). This systematic review was conducted according to the PRISMA methodology [26].

### 2.1. Study Search

We followed the PICOS methodology for the search strategy (Table 1). We performed the literature research using the string: (‘dentistry’/exp OR ‘dental medicine’ OR ‘dental system’ OR ‘dentistry’ OR ‘occupational dentistry’ OR ‘paediatric dentistry’ OR ‘paedodontics’ OR ‘pathology, oral’ OR ‘pediatric dentistry’ OR ‘pedodontics’ OR ‘practice, dental’ OR ‘specialties, dental’ OR ‘state dentistry’) AND ‘nitrous oxide’/exp for EMBASE; dent* AND ((nitrous oxide) AND Sedation) for PubMed; and “nitrous-oxide” AND dentistry for CENTRAL. Only studies published in English were included in the study screening database with no restriction to the time of publication.

### 2.2. Study Selection

After removing the duplicate articles from those retrieved with the abovementioned research, we listed the included studies using citation management software (Endnote VX9. Clarivate Analytics, Philadelphia, PA, USA). We included, as eligible studies, randomized controlled clinical trials, cross-sectional, case–control, and cohort studies published in English in peer-reviewed journals.

Two authors (FF and RG) independently performed an initial screening, based on the title and abstract only, evaluating the eligible studies. Then, the same two authors (FF and RG) further analyzed the full text of engaged articles for the final exclusion or inclusion in the systematic review and reported the reasons for exclusion of the articles that did not meet inclusion criteria. Another independent author (MF) helped retrieve the full-text version of some articles, resolved any disagreement on study inclusion, and supervised the whole procedure with a final check.

Each step of the research is reported and presented in a preferred reporting items for systematic reviews and meta-analyses (PRISMA) flow diagram (Figure 1).

### 2.3. Definition and Outcome

The population studied in this review included both adult and pediatric patients undergoing dental procedures.

For this study, we defined N_2_O alone as the use of only inhaled N_2_O–oxygen for dental sedation. All other sedative agents, placebo, or no drugs at all were considered as comparators.

As for the PROSPERO protocol, the main outcomes were patient satisfaction, recall, and procedure success. The additional outcomes were adverse events, perioperative pain intensity, opioid consumption, and surgeon’s satisfaction.

### 2.4. Data Extraction and Quality Assessment

Two authors (RG, FF) extracted data from the included studies using the Cochrane data collection form for intervention reviews for RCTs and non-RCTs independently. Two authors (RG, FF) assessed the methodological quality of the included studies using the Cochrane data collection form for intervention reviews for RCTs and non-RCTs and the Newcastle-Ottawa assessment scale (NOS) for case–control and cohort studies.

### 2.5. Data Analysis

Risk differences (RDs) were used as the meta-analytic measure of the association between N_2_O inhalation sedation and other medication. For each study, the proportion of patients used to calculate the RD and the corresponding 95% confidence interval (CI) using a 2 × 2 table was recorded. Heterogeneity between studies was assessed by using the Q statistic and I^2^, which is the proportion of total variance observed between the studies attributed to the differences between studies rather than to sampling error. I^2^ values of 25%, 50%, and 75% correspond to cut-off points for low, moderate, and high degrees of heterogeneity, and a *p* value of Q statistic less than 0.1 was considered significant [27]. If overall heterogeneity was significant, a random-effect model was used, otherwise, a fixed-effect model was used. Publication bias was assessed visually with funnel plots and with the Egger test [28]: a *p* value of less than 0.10 was considered significant. Data were analyzed using Stata version 11.2 (Stata Corp., College Station, TX, USA). All statistical tests were two-sided, and *p* values < 0.05 were regarded as usable in Section 3.

## 3. Results

### 3.1. Study Selection and Characteristics

A comprehensive search was performed from their inception until December 2024. Overall, we retrieved 1997 papers (1240 from Embase, 709 from PubMed, and 48 from CENTRAL), among which we removed 450 duplicates. Therefore, as shown in the flowchart, 1547 articles were identified as potentially relevant and screened, but subsequently among these we excluded 1395 articles by title and abstract. Hence, we performed a full-text analysis of 152 papers, of which 125 were excluded for the nine main reasons shown in the flowchart.

In total, after the full-text screening, we included the remaining 27 studies in the qualitative synthesis. Of these, 12 were randomized controlled trials (RCTs), 10 were cross-over trials, 4 were cohort studies, and one was a non-randomized controlled trial. A total of 13 of these studies were conducted in the UK, 3 in India, 3 in the USA, 2 in Brazil, 2 in Israel, and the remaining 4 were conducted in Netherlands, Japan, Canada, and Malaysia.

Eight studies evaluated patient satisfaction and enrolled a total of 422 patients: 165 were treated with N_2_O and 176 were treated with other medications (Table 2). Five studies evaluated the ability of the patient to recall tooth extraction and enrolled a total of 288 patients: 68 treated with N_2_O and 64 treated with other medications (Table 3). Three studies evaluated the successful completion of the dental procedure and included a total of 1578 patients: 434 were treated with N_2_O and 989 were treated with other medications (Table 4).

Only two studies evaluated adverse events as a primary outcome. The earliest, conducted by Wilson K.E., enrolled 70 patients, with 35 assigned to the N_2_O group [36]. The most recent study, by Soldani F., included 55 patients, 29 of whom received N_2_O [2]. Given the limited number of studies and the small sample sizes, a meta-analysis was not deemed appropriate.

### 3.2. Quantitative Synthesis

#### 3.2.1. Patient Satisfaction

The meta-analysis of the eight studies, involving 422 patients, evaluating patient satisfaction (Table 2) did not show significant differences in sedation with inhaled N_2_O for dental procedures compared to sedation with other medications (RD: −0.070; 95% CI [−0.20, 0.06]; *p*-value: 0.295), with significant heterogeneity among studies (*p* < 0.001, I^2^ = 88.7%). The forest plot is shown in Figure 2 and the funnel plot of the meta-analysis of the published studies is in the Appendix A.

#### 3.2.2. Ability to Recall

The meta-analysis of five studies, enrolling 288 patients, evaluating the ability of the patient to recall tooth extraction (Table 3) failed to show significant differences in sedation with inhaled N_2_O for dental procedures compared to sedation with other medications (RD: 0.02; 95% CI [−0.16, 0.20]; *p*-value: 0.828). The forest plot is shown in Figure 3 and the funnel plot of the meta-analysis of the published studies is in the Appendix A.

#### 3.2.3. Successful Completion of Dental Procedure

The meta-analysis of three studies (enrolling 1578 patients) evaluating successful completion of the dental procedure did not show significant differences in sedation with inhaled N_2_O for dental procedures compared to sedation with other medications (RD: −014; 95% CI [−0.41, 0.13]; *p*-value: 0.326). The forest plot is shown in Figure 4 and the funnel plot of the meta-analysis of the published studies is in the Appendix A.

## 4. Discussion

The meta-analysis did not show significant differences between N_2_O and other sedative agents in terms of patient satisfaction, perioperative memory recall, and successful completion of dental procedures. None of the meta-analyses yielded statistically significant differences, indicating that N_2_O offers a comparable clinical profile to alternative sedatives. In the first two outcomes explored (patient satisfaction and ability to recall) both adults and children were evaluated in the studies included in the pooled analysis. Otherwise, for the third outcome, only children were included. Unfortunately, we could not explore an outcome that we had set out to explore, which is the presence of adverse events; therefore, the choice of sedative in dental procedures cannot be recommended on such an important indication as the presence of adverse events.

A limitation of our meta-analysis is the variability of patient satisfaction evaluation in the different studies enrolled. We accepted the definition given by the authors, even though it differs between the studies. Unfortunately, there is no universally accepted definition of patient satisfaction. Another limitation of our meta-analysis is the limited number of patients enrolled for the ten studies that explored patient satisfaction (422 patients). For the second outcome (ability to recall) only five studies, enrolling 288 patients, were included. The largest number of patients (1578 patients) was explored for the third outcome (successful completion of dental procedure), but they were enrolled in only three studies, with great heterogeneity between studies. Despite limitations including heterogeneity, especially in patient satisfaction (I^2^ = 88.7%), among study designs and sedation protocols, as well as the relatively small sample size for some outcomes, the findings reinforce the role of N_2_O as an effective sedative agent in dental practice. Future standardized, high-quality RCTs are warranted to consolidate its comparative efficacy and safety across broader populations.

## 5. Conclusions

For all the outcomes (patient satisfaction, ability to recall, and successful completion of dental procedure) explored in this metanalysis there was no statistical difference between N_2_O and other sedative techniques. It was not possible to evaluate safety because it was not an outcome explored by the included studies. Future high-quality studies are needed to assess the safety of different sedative techniques.

## Figures and Tables

**Figure 1 medicina-61-00929-f001:**
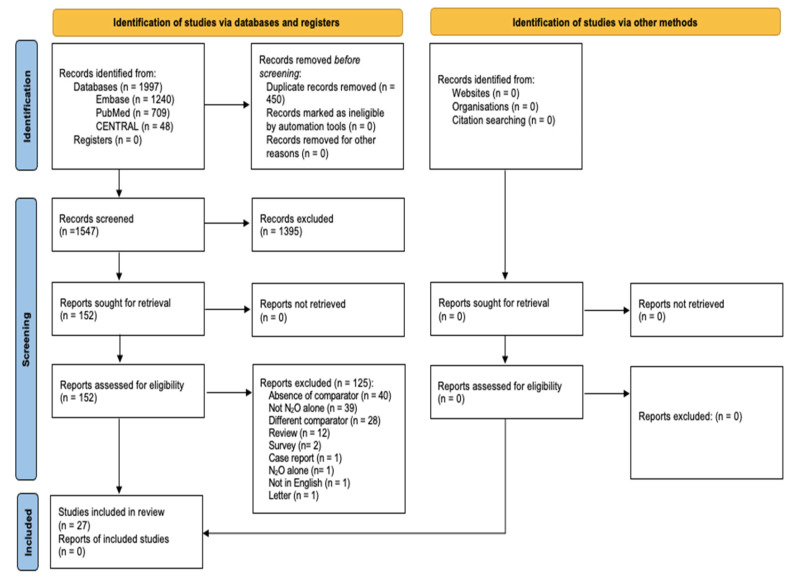
Preferred reporting items for systematic reviews and meta-analyses (PRISMA) flow diagram.

**Figure 2 medicina-61-00929-f002:**
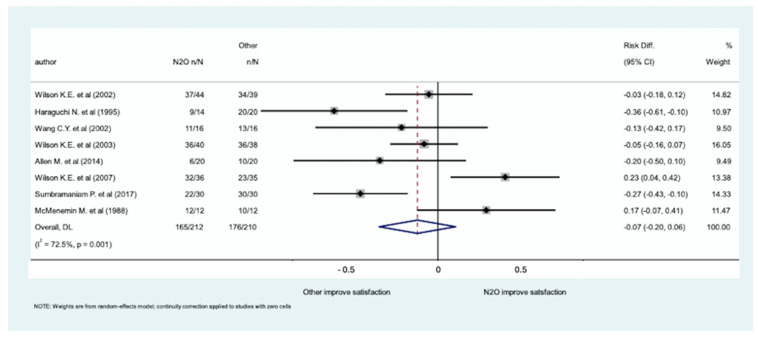
Forest plot of the eight studies, involving 422 patients, evaluating patient satisfaction [29,30,31,32,33,34,35,36].

**Figure 3 medicina-61-00929-f003:**
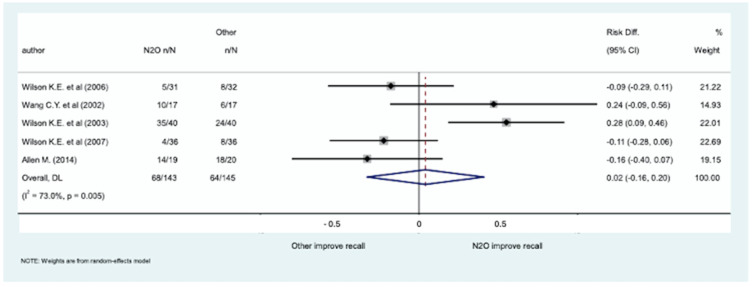
Forest plot of the 5 studies, involving 288 patients, evaluating the ability of the patient to recall tooth extraction [31,32,33,36,37].

**Figure 4 medicina-61-00929-f004:**
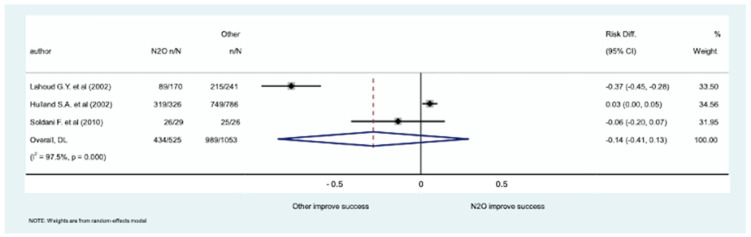
Forest plot of the 3 studies, involving 288 patients, evaluating the ability of the patient to recall tooth extraction [31,32,33,36,37].

**Table 1 medicina-61-00929-t001:** PICOS methodology for the search strategy.

Participants	Intervention	Comparison	Outcomes	Study Design
Adult and pediatric patients undergoing dental procedures	N_2_O inhalation sedation	Any sedative medication or placebo or no medication	Patient satisfaction; recall; success; adverse events	Cross-sectional, case–control, cohort, and RCTs

RCTs: randomized controlled trials; N_2_O: nitrous oxide.

**Table 2 medicina-61-00929-t002:** Patient satisfaction.

Author (Publ. Year) [Ref.]	N. of Patients Enrolled	N. of Patients Treated with N_2_O	N. of Patients Treated with Other Medications	Other Medication	Study Design	Adults/ Children	Patient Satisfaction Evaluation
Wilson K.E. et al. (2002) [29]	83	44	39	Oral midazolam	RCT crossover	Children	Questionnaire: pleasant, acceptable, and unpleasant
Haraguchi N. et al. (1995) [30]	34	14	20	Sevoflurane	RCT	Adults	Questionnaire: odour, general opinion, and would have same inhalation again
Wang C.Y. et al. (2002) [31]	32	16	16	Sevoflurane	RCT crossover	Adults	Patient intraoperative feeling VAS (0 = pleasant; 10 = unpleasant)
Wilson K.E. et al. (2003) [29]	78	40	38	Intravenous midazolam	RCT crossover	Children	Questionnaire: pleasant, acceptable, and unpleasant; preferred type of sedation
Allen M. et al. (2014) [32]	40	20	20	Sevoflurane	RCT	Adults	Questionnaire: pleasant and not pleasant
Wilson K.E. et al. (2007) [33]	71	36	35	Sublingual midazolam	RCT crossover	Adolescents	Questionnaire: how they felt about sedarion, what they liked or disliked, and preferred type of sedation
Sumbramaniam P. et al. (2017) [34]	60	30	30	Oral triclofos sodium	RCT	Children	Patient’s acceptance of route of administration
McMenemin M. et al. (1988) [35]	24	12	12	Isoflurane	Prospective crossover trial	Adults	VAS scale of level of sedation and mood; pleasant or unpleasant

Publ.: publication; Ref.: reference; RCT: randomized controlled trial; VAS: visual analogue scale.

**Table 3 medicina-61-00929-t003:** Ability to recall extraction.

Author (Publ. Year) [Ref.]	N. of Patients Enrolled	N. of Patients Treated with N_2_O	N. of Patients Treated with Other Medications	Other Medication	Study Design	Adults/ Children
Wilson K.E. et al. (2006) [36]	63	31	32	Oral midazolam	RCT crossover	Children
Wang C.Y. et al. (2002) [31]	34	17	17	Sevoflurane	RCT crossover	Adults
Wilson K.E. et al. (2003) [37]	80	40	40	Intravenous midazolam	RCT crossover	Children
Wilson K.E. et al. (2007) [33]	72	36	36	Sublingual midazolam	RCT crossover	Adolescents
Allen M. (2014) [32]	39	19	20	Sevoflurane	RCT	Adults

Publ.: publication; Ref.: reference; RCT: randomized controlled trial.

**Table 4 medicina-61-00929-t004:** Successful completion of dental procedure.

Author (Publ. Year) [Ref.]	N. of Patients Enrolled	N. of Patients Treated with N_2_O	N. of Patients Treated with Other Medications	Other Medication	Study Design	Adults/ Children
Lahoud G.Y. et al. (2002) [38]	411	170	241	Sevoflurane + N_2_O	RCT	Children
Hulland S.A. et al. (2002) [39]	1112	326	786	Oral midazolam	RCT	Children
Soldani F. et al. (2010) [2]	55	29	26	N_2_O + sevoflurane	RCT crossover	Children

Publ.: publication; Ref.: reference; RCT: randomized controlled trial.

## Data Availability

Upon reasonable request, the corresponding author (M.F.) can provide the data that supports the findings of this systematic review with meta-analysis.

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
