# Peer review of "Efficacy and Safety of Nitrous Oxide (N2O) Inhalation Sedation Compared to Other Sedative Agents in Dental Procedures: A Systematic Review with Meta-Analysis"

_medicina, 2025, doi:10.3390/medicina61050929_

Round 1
Reviewer 1 Report
Comments and Suggestions for Authors
Dear authors
I belive that for this final form of the article you worked a lot .However my opinion is that it would need some adjustments.
Introduction is good and brings qualitative information related to the topic.
Study selection-my opinios is that you should remake the prisma chart because has some missing informations.
Results- related to Patients satisfaction my opinion is that you should expose how was this ‘’Patient satisfaction” evaluated in the 8 studies.
Discussion is too brief in relation with the results obtained and must be extended.
Author Response
COMMENTS 1“Dear authors
I belive that for this final form of the article you worked a lot .However my opinion is that it would need some adjustments.
Introduction is good and brings qualitative information related to the topic.
Study selection-my opinios is that you should remake the prisma chart because has some missing informations.”
RESPONSE 1 : we have updated the flow chart according to the “PRISMA 2020 flow diagram for new systematic reviews which included searches of databases, registers and other sources”, downloaded at https://www.prisma-statement.org/prisma-2020-flow-diagram (lines 136-138)
COMMENTS 2 “Results- related to Patients satisfaction my opinion is that you should expose how was this ‘’Patient satisfaction” evaluated in the 8 studies.”
RESPONSE 2: According to your comment, we have modified Table 1, adding one more column explicating how patient satisfaction was investigated (lines 194-195).
COMMENTS 3“Discussion is too brief in relation with the results obtained and must be extended.”
RESPONSE 3: We extended the discussion as suggested (lines 254-257, 263-265, 271-275)
Reviewer 2 Report
Comments and Suggestions for Authors
The manuscript presents a timely and well-organized systematic review and meta-analysis evaluating the efficacy and safety of nitrous oxide (Nâ‚‚O) in dental procedures compared to other sedative techniques. The subject is of high clinical relevance, especially considering the increasing demand for minimally invasive and patient-centered sedation strategies in dentistry. The review is appropriately registered (PROSPERO: CRD42020213429) and follows PRISMA guidelines, enhancing its transparency and reproducibility. Including both adult and pediatric populations broadens the scope and applicability of the findings.
The methodology is comprehensive, covering multiple databases and a detailed PICOS framework. However, the authors should justify their selection of outcome measures more explicitly, particularly the exclusion of adverse events as a primary outcome, given its importance in safety assessment. Although the search strategy appears thorough, the flow diagram lacks detailed citations for excluded studies, which is essential for replicability. It is recommended that a supplementary table listing these excluded articles and their reasons be added. Additionally, table abbreviations should be clearly defined, as the lack of caption details may hinder reader comprehension.
The statistical analysis is well-conducted, with appropriate use of random/fixed-effects models based on heterogeneity levels. Yet, publication bias assessment using funnel plots and Egger’s test is mentioned but not adequately discussed in the main text. Furthermore, while the results show no significant difference between Nâ‚‚O and comparators across outcomes, the heterogeneity—especially in patient satisfaction (I² = 88.7%)—suggests variation in study populations or methodologies that should be explored in greater depth in the discussion.
The findings support the conclusions, but the manuscript would benefit from a more critical reflection on the limitations, including the small number of high-quality RCTs and inconsistent reporting of safety outcomes.
Overall, this study contributes meaningful evidence to the debate over sedation choices in dental care. With some clarifications and minor revisions, it holds strong potential for publication.
Author Response
COMMENTS 1 “The manuscript presents a timely and well-organized systematic review and meta-analysis evaluating the efficacy and safety of nitrous oxide (Nâ‚‚O) in dental procedures compared to other sedative techniques. The subject is of high clinical relevance, especially considering the increasing demand for minimally invasive and patient-centered sedation strategies in dentistry. The review is appropriately registered (PROSPERO: CRD42020213429) and follows PRISMA guidelines, enhancing its transparency and reproducibility. Including both adult and pediatric populations broadens the scope and applicability of the findings.
The methodology is comprehensive, covering multiple databases and a detailed PICOS framework. However, the authors should justify their selection of outcome measures more explicitly, particularly the exclusion of adverse events as a primary outcome, given its importance in safety assessment.”
RESPONSE 1: Only two studies evaluated adverse events as a primary outcome. The earliest, conducted by Wilson K.E., enrolled 70 patients, with 35 assigned to the Nâ‚‚O group [36]. The most recent study, by Soldani F., included 55 patients, 29 of whom received Nâ‚‚O [2]. Given the limited number of studies and the small sample sizes, a meta-analysis was not deemed appropriate (lines 204-208).
COMMENTS 2 “Although the search strategy appears thorough, the flow diagram lacks detailed citations for excluded studies, which is essential for replicability. It is recommended that a supplementary table listing these excluded articles and their reasons be added.”
RESPONSE 2: According to your comment, we have added a supplementary table listing all the excluded articles, by reason of exclusion and detailed citations for each of them (Table S1).
COMMENTS 3“Additionally, table abbreviations should be clearly defined, as the lack of caption details may hinder reader comprehension.”
RESPONSE 3: We have modified the tables in the article accordingly, clarifying all the abbreviations.
COMMENTS 4 “The statistical analysis is well-conducted, with appropriate use of random/fixed-effects models based on heterogeneity levels. Yet, publication bias assessment using funnel plots and Egger’s test is mentioned but not adequately discussed in the main text. Furthermore, while the results show no significant difference between Nâ‚‚O and comparators across outcomes, the heterogeneity—especially in patient satisfaction (I² = 88.7%)—suggests variation in study populations or methodologies that should be explored in greater depth in the discussion.”
RESPONSE 4: We extended the discussion as suggested (lines 254-257, 263-265)
COMMENTS 5 “The findings support the conclusions, but the manuscript would benefit from a more critical reflection on the limitations, including the small number of high-quality RCTs and inconsistent reporting of safety outcomes.”
RESPONSE 5: We extended the discussion as suggested (lines 271-275)
COMMENTS 6 “Overall, this study contributes meaningful evidence to the debate over sedation choices in dental care. With some clarifications and minor revisions, it holds strong potential for publication.”
RESPONSE 6: Thank you for your review and appreciation.